

# Tracking the spread of a passive tracer through Southern Ocean water masses

Jan D. Zika[1], Jean-Baptiste Sallée[2], Andrew Meijers[3], Alberto Naveira-Garabato[4], Andrew J. Watson[5], Marie-Jose Messias[5], and Brian  King[6]

[1]School of Mathematics and Statistics. University of New South Wales, Sydney, Australia
[2]Sorbonne Université, CNRS, LOCEAN Laboratory, Paris, France
[3]British Antarctic Survey, Cambridge, UK
[4]University of Southampton, National Oceanography Centre, UK
[5]University of Exeter, UK
[6]National Oceanography Centre, Southampton, UK

**Correspondence:** Jan D. Zika (J.Zika@unsw.edu.au)

**Abstract.** A dynamically passive inert tracer was released in the interior South Pacific Ocean at latitudes of the Antarctic Circumpolar Current. Observational cross sections of the tracer were taken over four consecutive years as it drifted through Drake Passage and into the Atlantic Ocean. The tracer was released within a region of high salinity relative to surrounding waters at the same density. In the absence of irreversible mixing a tracer remains at constant salinity and temperature on an

isopycnal surface. To investigate the process of irreversible mixing we analysed the tracer in potential density versus salinity-anomaly coordinates. Observations of high tracer concentration tended to be collocated with isopycnal salinity anomalies. With time an initially narrow peak in tracer concentration as a function of salinity at constant density, broadened with the tracer being found at ever fresher salinities, consistent with diffusion-like behaviour in that coordinate system. The second moment of the tracer as a function of salinity suggested an initial period of slow spreading for approximately 2 years in the Pacific, followed

by more rapid spreading as the tracer entered Drake Passage and the Scotia Sea. Analysis of isopycnal salinity gradients based on the Argo programme suggests that part of this apparent change can be explained by changes in background salinity gradients while part of the change may be explained by geographical changes in background mixing.

## 1 Introduction

Isopycnal mixing is a key controller of the transport of carbon (Sallée et al., 2012; Gnanadesikan et al., 2015) and heat (Gregory,

2000) into the deep sea, and of the stability of the long-term climate system (Sijp et al., 2006). The influence of isopycnal mixing is nowhere more prevalent than in the Southern Ocean, where steeply sloping isopycnals form a connection between the atmosphere and the deep ocean (Marshall and Speer, 2012). Strong gradients of temperature and salinity along isopycnals imply a necessary balance between advection of relatively warm-saline Circumpolar Deep Water and its freshening by isopycnal and diapycnal mixing (Zika et al., 2009; Naveira Garabato et al., 2016). Here we explore how a passive tracer released as part of

the Diapycnal and Isopycnal Mixing Experiment in the Southern Ocean (DIMES; dimes.ucsd.edu) project mixes through this background salinity and temperature gradient.



Previous tracer release experiments have found that tracers closely followed the release isopycnal surface, spreading slowly across isopycnal surfaces (i.e. across vertically stacked density layers, Ledwell et al., 1993; Morris et al., 2001; Ledwell et al., 2011; Watson et al., 2013). Tracer profiles in these studies were typically averaged over sections as a function of density and then transformed into depth coordinates through a reference density vs depth profile. The motivation for averaging at constant density is that the tracer can move to different depth levels through adiabatic reversible motions. Moving to different density values in the ocean interior (i.e. away from sources and sinks of heat and salt) requires irreversible mixing, which either directly mixes the tracer across isopycnals or causes diapycnal advection (e.g. Toole and McDougall, 2001).

Just as vertical displacements of the tracer can occur without diapycnal mixing, lateral spreading of the tracer can occur without any irreversible mixing. The tracer can be spread out or contract geographically due to diverging or converging ocean currents, respectively, and through rearrangement by mesoscale flows. This motivates the use of coordinates which preserve adiabatic horizontal/isopycnal motions just as isopycnal coordinates preserve adiabatic vertical motions.

Various cross-stream coordinates have been proposed, such as sea surface height (Sokolov and Rintoul, 2009; Meijers et al., 2011), dynamic height (Naveira Garabato et al., 2011) and density on pressure surfaces (Zika et al., 2013). However, these are not necessarily conserved following the tracer path and implicitly assume that the flow is 'equivalent-barotropic' throughout the entire water column (Killworth, 1992). Any subtle spiralling of the velocity vector with depth can accumulate to create a mismatch between the flow direction on a particular isopycnal and the tangent lines of SSH or dynamic height contours at the surface.

It has long been understood that, in the absence of irreversible mixing, a water parcel will remain at constant salinity and temperature (Iselin, 1939). Since the mixing of water parcels with the same density is thought to dominate over mixing at different densities, a variable known as 'spice' is often used to track along-isopycnal motion. Under the assumption of a linear equation of state, spice describes the density-compensated variations in temperature and salinity along a constant density surface, and is orthogonal to density in the temperature - salinity plane (Veronis, 1972). The spice variable has been exploited by a number of authors (e.g. Rudnick and Martin, 2002). Here we simply use salinity at constant density as our coordinate which is equivalent locally to a typical definition of 'spice'.

Here, we have analysed tracer data collected during DIMES using an isopycnal-salinity coordinate with the aim of helping to understand isopycnal dispersion and complimenting previous studies that have used more conventional approaches to estimate isopycnal mixing rates. Using these data Tulloch et al. (2014) estimated the isopycnal diffusion coefficient to be $710 \pm 260$ $\mathrm{m^2\,s^{-1}}$ along the Antarctic Circumpolar Current (ACC) in the South Pacific Ocean near the release depth of 1500 m. Those authors applied the ensemble tracer patch area approach (Garrett, 1983). In a parallel study, LaCasce et al. (2013) analyzed data from sub-surface Lagrangian drifters released during the DIMES campaign. They estimated the isopycnal diffusivity to be $800 \pm 200$ $\mathrm{m^2\,s^{-1}}$ using the Lagrangian dispersion method (Garrett, 1983).

We show, in Section 2, that the tracer released in the interior South Pacific spreads geographically in an inhomogeneous way as it follows the ACC and is stirred by eddies. In Section 3, we show that some of this inhomogeneity is reduced when the tracer is projected into salinity coordinates. In Section 4, we project the spreading of the tracer in salinity coordinates into equivalent geographical coordinates. We then attempt to reconcile the growth of its second moments in equivalent geographical





coordinates in terms of diffusion coefficients in Section 5, and we discuss these results in terms of existing theories in Section 6. Concluding remarks are given in Section 7.

## 2 Tracer data

In February 2009, 76 kg of the inert tracer trifluoromethyl sulphur pentafluoride ($CF_3SF_5$) were released at the potential density (referenced to 1000 m) of 32.325 kg m$^{-3}$ (neutral density $\approx$ 27.9 kg m$^{-3}$; depth $\approx$ 1500 m) near 58°S, 253°E. The tracer was chosen as it has a very low background atmospheric concentration and nearly negligible interior ocean concentration, is non-toxic and is chemically inert in the environment and is detectable at extreme dilution (Ho et al., 2008). The tracer was released in a cross pattern approximately 20 km wide, between the Subantarctic and Polar fronts and at the density of Upper Circumpolar Deep Water. The initial tracer distribution, determined by sampling within two weeks of the release, was found to be centered about 4 meters below the target density surface. The rms spread of the tracer in density was documented as approximately 0.0015 kg m$^{-3}$, corresponding to a vertical rms spread of 5.5 m. For details of the release method and measurement technique, see Ledwell et al. (2011).

Several subsequent surveys were undertaken: in February 2010 in the region 57°S to 62°S and 255°E to 275°E (Fig. 1a); in December 2010 (Fig. 1b), April 2011 (Fig. 1c) and August 2011 (Fig. 1d) in the regions west of and at Drake Passage; and along the western and northern margins of the Scotia Sea in 2012 (February-March, Fig. 1e) and 2013 (March-April, Fig. 1f) to measure the change in distribution of the tracer with time. Tracer sampling and shipboard chemical analysis are described in Ledwell et al. (2011) and in Watson et al. (2013).

We have analysed 11 sections taken at 5 locations, here named: South Pacific (actually a regional survey), South-East Pacific (along 282°E), Drake Entry (near 293°E), Drake Exit (along the section commonly termed SR1b near 303°E), and North Scotia (along the North Scotia Ridge). These locations are indicated in Fig. 1.

Year-on-year, the tracer was found increasingly toward the east and in lower concentrations (Fig.2; Watson et al., 2013). The tracer closely followed the release isopycnal surface, spreading slowly across isopycnal surfaces (i.e. across vertically stacked density layers). Tracer profiles averaged over sections as a function of density and then transformed into depth coordinates through a reference density/depth profile were approximately Gaussian. The spread of these Gaussian profiles with time and/or with distance to the east, yielded diapycnal diffusivities of O(2 x 10$^{-5}$) m$^2$ s$^{-1}$ in the South Pacific (Ledwell et al., 2011; Watson et al., 2013), and O(3 x 10$^{-4}$) m$^2$ s$^{-1}$ in the Scotia Sea, where the ACC flows over comparatively shallow and complex topography (Watson et al., 2013).

The tracer was not always delimited at the northern and southern ends of the transects or surveys due to compromises dictated by ship time limitations. Sampling was planned with information on the locations of the Polar Front to the south and the Subantarctic Front (west of Drake Passage) or of the continental slope (east of Drake Passage) to the north. The tracer had been released midway between the two fronts, and so when compromises in sampling were needed, sampling was curtailed beyond the fronts, especially north of the Subantarctic front in the Pacific, guided by the notion that the fronts are barriers to cross-ACC transport (e.g. Naveira Garabato et al., 2016) and also by an altimetry-based prediction of tracer patch spreading. It

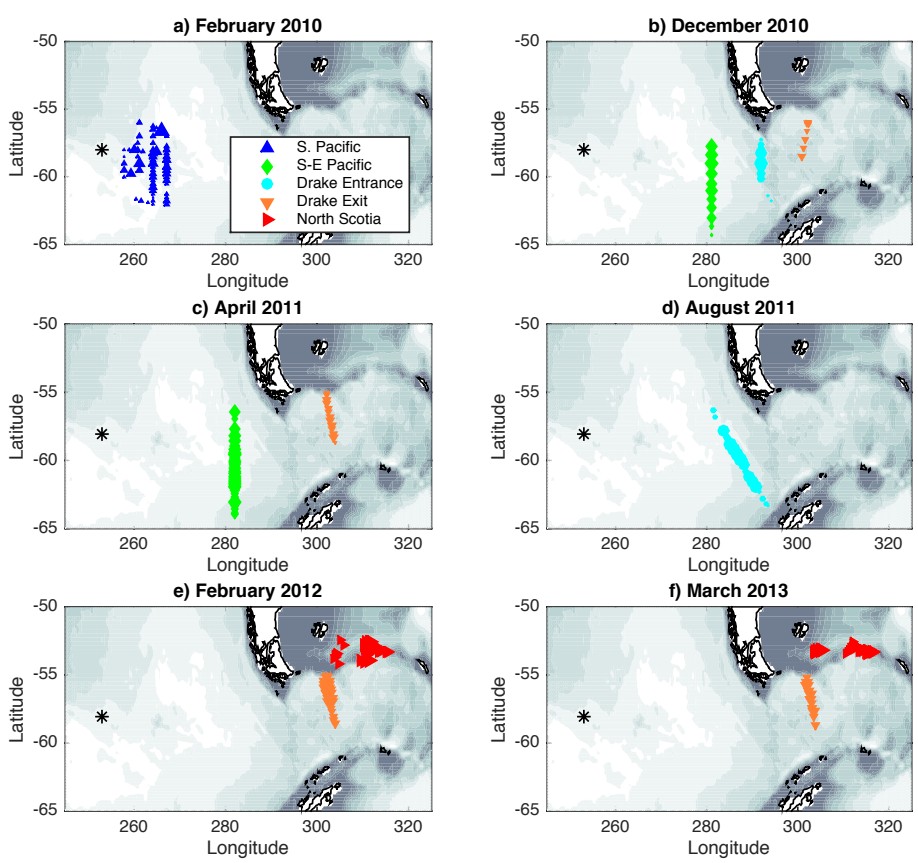

**Figure 1.** Cast locations for each summer voyage in which tracer measurements were taken. The size of each marker corresponds to the peak tracer concentration for that cast. Marker sizes are scaled relative to the largest measured tracer concentration for that time period (i.e. markers corresponding to the cast where the largest concentration was measured have the same size between all six figures; see Fig.2 for maximum concentrations for each section). The star indicates the release location, upward triangles the South Pacific region (US Voyage 2), diamonds the South-East Pacific section (along 282°E), circles the Drake Entry section (along Phoenix Ridge), downward triangles Drake Exit section (otherwise known as SR1b) and right triangles the North Scotia section.



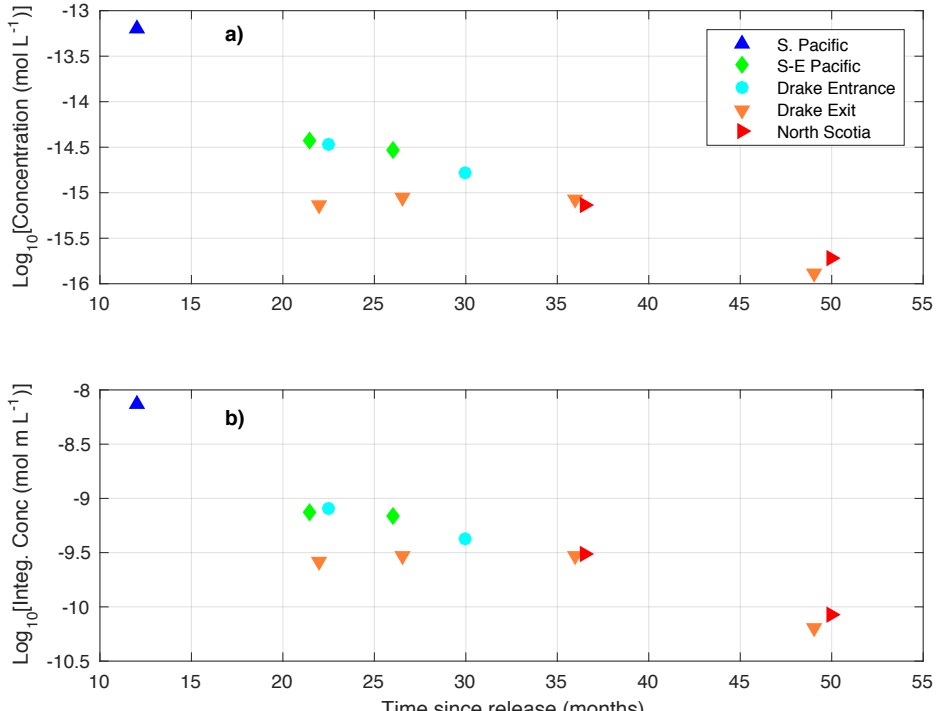

**Figure 2.** a) The peak concentration measured ($\log_{10}$ mol L$^{-1}$) at each section as a function of time since release (months). b) Peak depth-integrated concentration ($\log_{10}$ mol m L$^{-1}$). Although higher concentrations could be inhomogeneously distributed either side of the sections at a given time, the time series suggests that the concentration peak is closer to the eastern South Pacific before 30 months, and closer to or beyond the North Scotia Ridge by the last transect near 50 months

turned out that little tracer was detected south of the Polar Front when time allowed for sampling there. However, sections in the Pacific often found high concentrations in the northernmost (although still south of the Sub Antarctic Front) stations, suggesting that there was tracer to the north of the sampled area. In fact, the evidence is that not very much of the tracer spread to the north of the Subantarctic Front. The DIMES float trajectories, shown in Fig. 3 of LaCasce et al. (2013), illustrate that virtually none of the floats went far enough north to avoid transiting through Drake Passage. The ensemble of numerical simulations

for realistic conditions, calibrated with the available tracer observations, shown in Fig. 1 of Tulloch et al. (2014) also suggest that relatively little tracer was missed to the north or south of the sampled regions. Hence, we argue that tracer sampling was sufficiently representative to support our main conclusions, though the shortfall of sampling adds to the uncertainty of our quantitative analysis.

      The tracer was apparently distributed over a smaller meridional range at the Drake Passage Exit section than in the South-East

Pacific. This could be due to a decrease in meridional spread as the tracer moves downstream associated with the contraction of the ACC as it flows through Drake Passage or due to sampling issues. In either case it poses a problem to conventional analysis of cross-ACC dispersion in terms of a Fickian eddy diffusivity since the second moment in geographical space decreases with





time. We shall see shortly that analysis of the along-isopycnal spreading of the tracer into waters with different salinities helps
to circumvent this problem.

## 3  Tracer dispersion in density versus salinity-anomaly coordinates

The tracer was found to spread inhomogeneously in the horizontal. Figure 3 shows the distribution of the tracer as a function
of latitude and depth (panels a-c) and latitude and density (d-f) and in temperature and salinity coordinates (g-i) for the South
Pacific section in February 2010 (a,d and g), South-East Pacific section in December 2010 (b, e and h) and the Drake Exit
section in March 2011 (c, f and i).

The tracer spread isopycnally and formed multiple peaks in concentration, suggesting that the patch had been teased into
filaments. When the tracer distribution is projected onto temperature versus salinity coordinates, these individual maxima are
not seen, suggesting that filaments preserve their temperature and salinity values. To investigate this effect further, we show
both the salinity and tracer concentration on the release isopycnal (Fig.3 j-l). These two variables are also plotted against one
another as a scatter plot (Fig.3 m-o). Two to three years into the experiment (i.e. during the late 2010 and 2011 voyages), higher
tracer concentrations are found where the highest salinities are measured on the isopycnal. The core of the tracer is apparently
moving with or close to the salinity maximum on the isopycnal.

On an isopycnal surface, the isopycnal salinity anomaly ($S'$) is defined as the local salinity minus some reference salinity
($S_0$). We choose the reference salinity to be the salinity at the location on the isopycnal where the tracer concentration is largest.
On the release isopycnal $S_0$ remains close to 34.65g kg$^{-1}$ throughout our study domain, decreasing slightly toward the east,
but is always larger than 34.645g kg$^{-1}$. In order for the tracer to disperse in this coordinate, it must either cross isotherms at
constant density (i.e. with a compensating change in temperature) or cross isopycnal surfaces. Our focus here is on dispersion
in the isopycnal direction.

The choice to define $S_0$ separately for each section, rather than as the salinity value on which the tracer was injected, is
intended to account for migration of the peak due to gradual along-isopycnal transport. However, the effect of this choice is
negligible relative to other sources of uncertainty.

In Figure 4, the tracer distributions for each section are mapped onto density versus salinity anomaly coordinates. While in
geographical coordinates the dispersion of the tracer has multiple maxima and minima and appears to contract meridionally
with time, in density versus salinity anomaly coordinates the tracer spreads out more monotonically.

In order to quantify the spreading of the tracer in isopycnal salinity anomaly coordinates, we apply a nonlinear least squares
fit (using Matlab's fminsearch algorithm) of a two-dimensional Gaussian distribution in that reference frame. This is done at
each section to yield the standard deviation of salinity in the along-isopycnal direction. As mentioned above, growth of the
tracer patch in the diapycnal direction during the initial two years of the experiment was analyzed by Ledwell et al. (2011) and
by Watson et al. (2013) and further discussion of diapycnal mixing is left to future work.

The evolution of the second moment of the tracer in isopycnal salinity anomaly coordinates (the square of the standard
deviation) is shown in Fig. 5. There is an apparent weak spreading of the tracer in the first 24 months, with the tracer patch

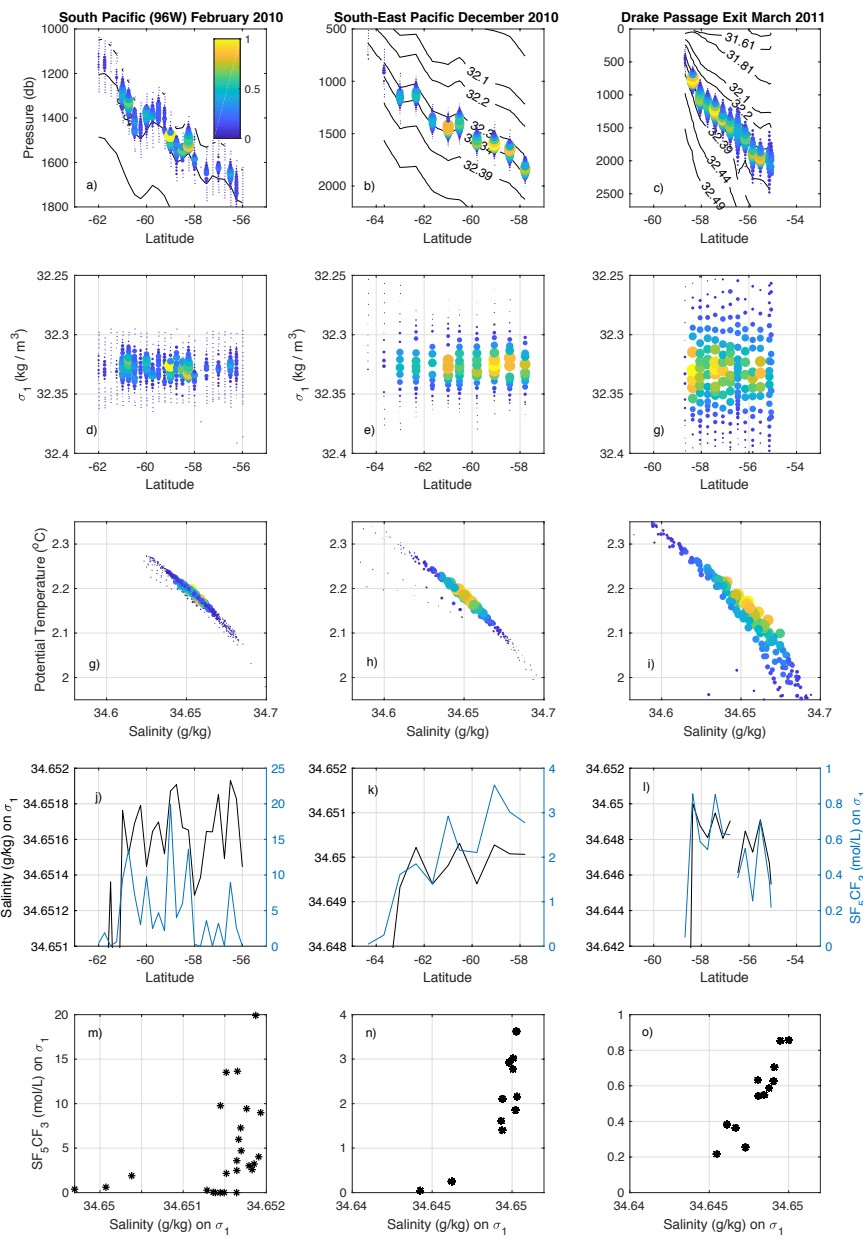

**Figure 3.** a-c) Depth and latitude of tracer measurements, with contours of potential density referenced to 1000m ($\sigma_0$; not evenly spaced). The Polar Front and the Subantarctic Front, respectively, are where the slope of the density surfaces is large, near the southern and northern ends of these sections. d-f) Potential density and latitude of tracer measurements. g-i) Potential temperature and salinity. j-l) Salinity and tracer on the $\sigma_1$=32.325 kg m$^{-3}$ surface. m-o) Scatter plot of salinity and tracer concentration on the $\sigma_1$=32.325 kg m$^{-3}$ surface. For panels a-i, colour represents tracer concentration relative to the peak measured for that section. Panels a,d and g are from the South Pacific cruise of February 2010; b, e and h the South-East Pacific section of December 2010; c, f and i the Drake Exit section of March 2011.



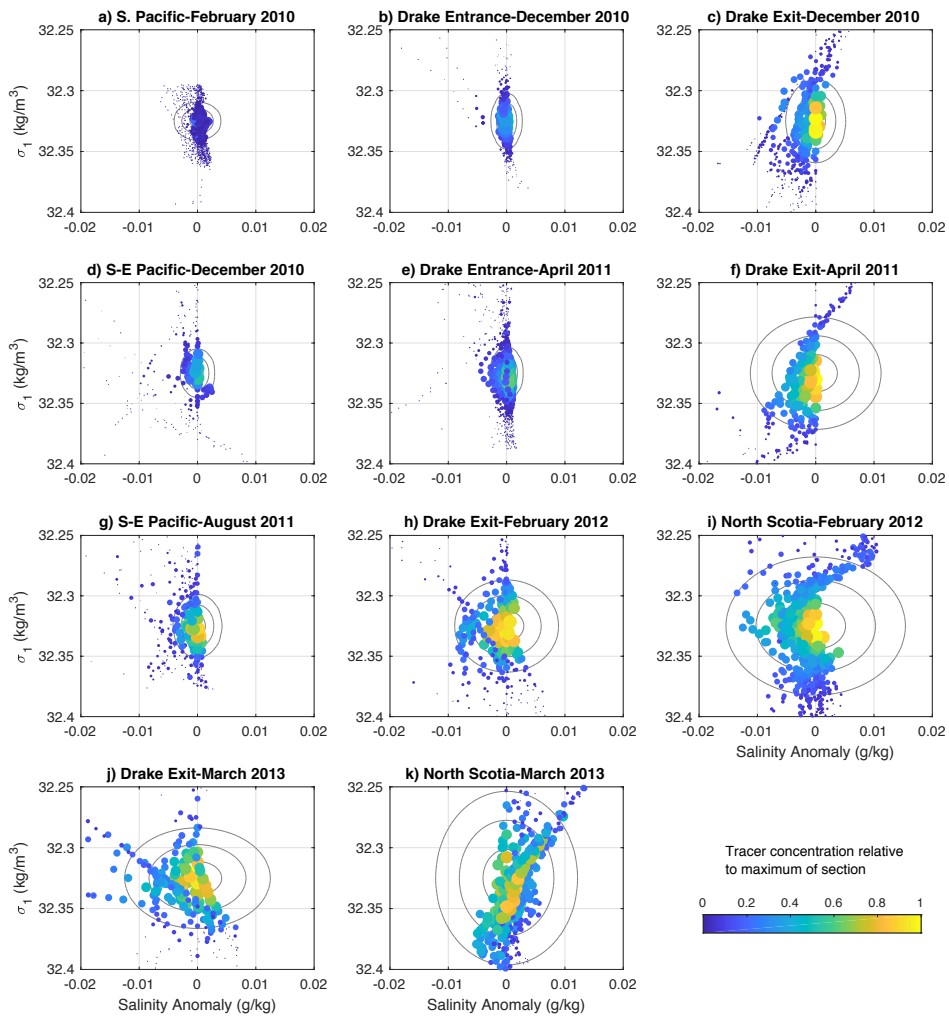

**Figure 4.** Data from all 11 sections from the five DIMES cruises discussed are shown in potential density versus isopycnal salinity anomaly coordinates. Each point is colored by the tracer concentration relative to the maximum of that section (see Fig.2 for these values). Grey contours show the $\sigma/2$, $\sigma$ and $3\sigma/2$ contours of a fit of a 2D Gaussian curve to these data.

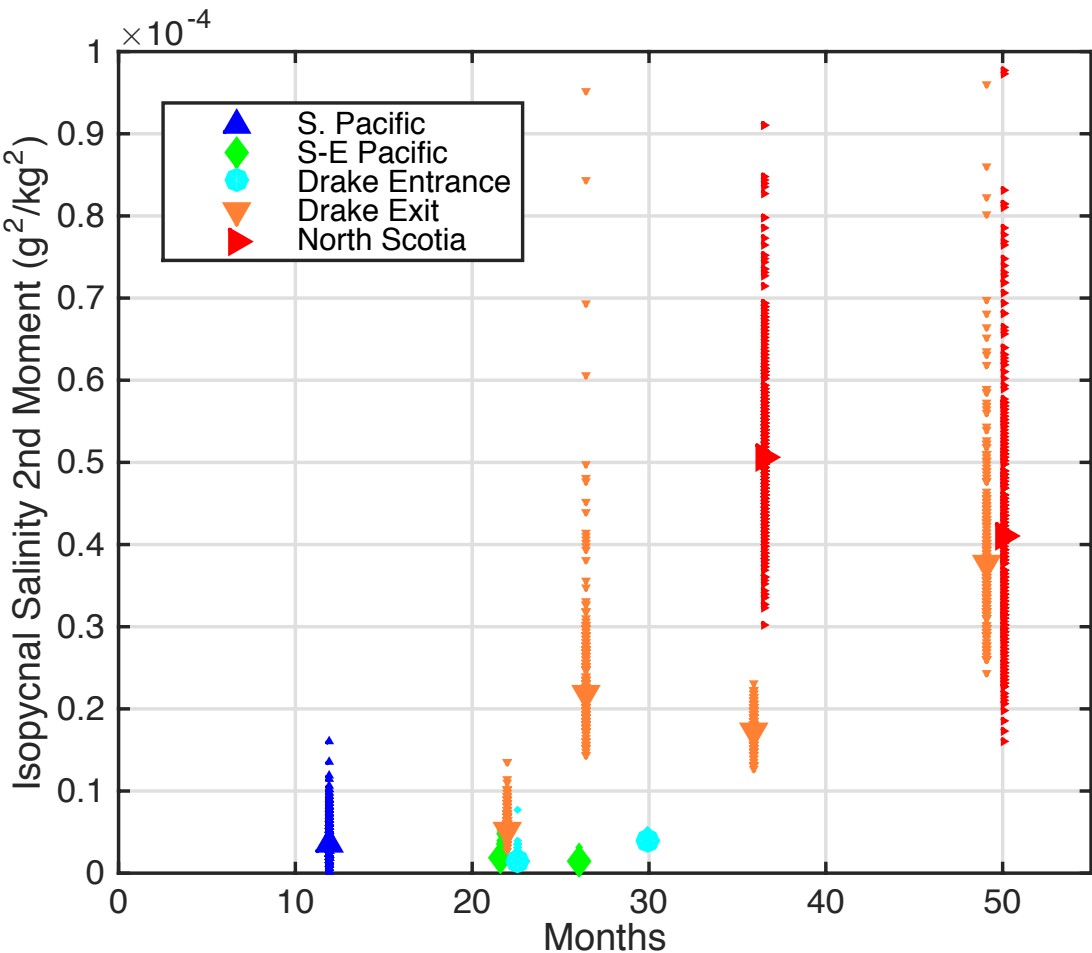

**Figure 5.** Isopycnal second moment of the tracer in salinity coordinates derived from the Gaussian fits in Fig. 3 versus time since release. Each small marker (making up the error bars) represents a single member of the bootstrap ensemble, and the larger marker the result of using all the data.





growing to second moment values generally less than $5 \times 10^{-6}$ (g/kg)$^2$; then, over the subsequent 24 months, more rapid spreading of the tracer occurs, reaching second moment values of order $5 \times 10^{-5}$ (g/kg)$^2$.

To account for the various sources of sampling error, we estimate uncertainty using the following bootstrapping method. For each section, the tracer observations are randomly sub-sampled allowing for repeated sampling of the same data. For a section with $N$ samples, $N$ observations are chosen at random (with repetition), and the second moment determined . This is then repeated $N$ times. Each estimate is shown in Fig.5.

The uncertainty analysis reveals a larger range of second moment estimates for the initial South Pacific survey after 12 months than in surveys conducted after 20-30 months in the same region, potentially indicating an initial lack of correspondence between salinity and tracer filaments (Fig. 5). Uncertainties then increase for the later period and in the Scotia Sea, likely due to reduced tracer concentrations, poor coverage of the Gaussian distribution, and a lack of consistency of the Gaussian model due to anisotropic variations in mixing and a changing background salinity and density field.

Nonetheless, there is a notable increase in the rate at which the tracer spreads in isopycnal salinity anomaly coordinates between the initial two years and the subsequent years of the experiment. This could be explained by a change in the rate at which the tracer is mixed, or by a change in background salinity gradient. In the next section, we investigate the these explanations.

## 4 Projection from isopycnal salinity anomaly to distance coordinates

In order to interpret our analysis in terms of geographical dispersion, we next relate the isopycnal salinity anomaly coordinate to an equivalent cross-stream distance. We use the climatological mean distance between isohalines along the ACC to effect this transformation. Although such a transformation results in information loss relating to the moving salinity coordinate, it is instructive in linking dispersion to an apparent mixing coefficient for comparison with other work.

Distances between isohalines are determined using the entirety of available hydrographic data for the region. The hydrographic compilation used in this study includes ship-based observations from the World Ocean Database (http://www.nodc.noaa.gov/OC5/SELECT /dbsearch/dbsearch.html) and from the Argo program (http://www.argo.ucsd.edu, http://argo. jcommops.org), as well as from ship-based observations directly obtained from individual PIs (see Sallée et al., 2010).

For each vertical cast, potential density and salinity are determined. Salinity is then linearly interpolated onto the $\sigma_1 = 32.325$ kg m$^{-3}$ isopycnal surface (Fig. 7). Maps of these raw salinity values show a clear ridge of high salinity running along the ACC on this isopycnal. Higher salinity gradients are apparent further to the east as the ACC contracts. It is along this ridge that the tracer is released.

If fine-resolution (e.g., eddy-resolving), real-time and realistic salinity data were available on the isopycnal surface, we would form a time-varying salinity coordinate by calculating the area contained between salinity ranges and remap this onto a quasi-meridional distance (e.g., following Marshall et al., 2006). However, such a density of data is not available from an observational product. Instead, we estimate the distance coordinate directly from the in-situ data. We do this by, first, estimating





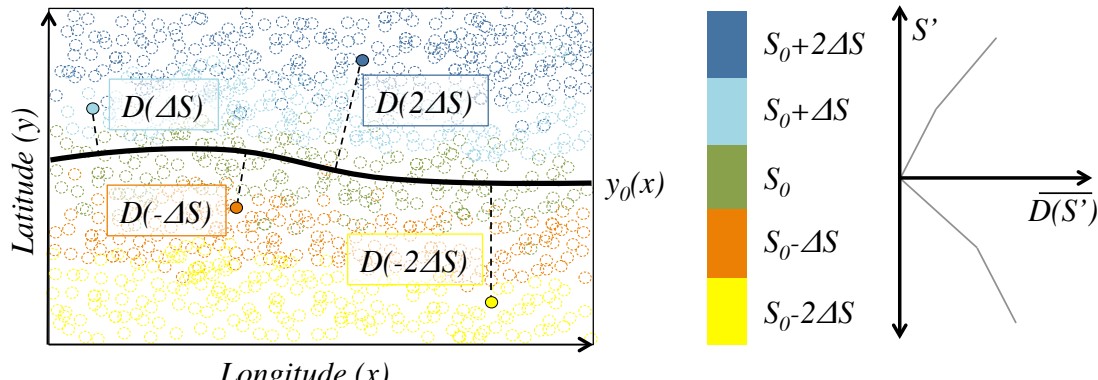

**Figure 6.** Schematic describing the method by which the mean distance as a function of salinity anomaly $(\overline{D(S')})$ is calculated. Circles represent salinity observations on an isopycnal surface. The latitude of observations close to the target salinity ($S_0 \pm \Delta S/2$; green) are used to define a locus of latitudes ($y_0(x)$; solid line). The minimum great circle distance ($D$) of each observation from $y_0(x)$ is then determined and binned as a function of the salinity anomaly ($S' = S - S_0$). Distances are averaged within each bin, giving a relationship between salinity anomaly and distance $\overline{D(S')}$.

the mean location of the salinity contour along which the tracer is centred; and, second, calculating the average distance from that contour to other salinity measurement sites.

The western South Pacific and South Atlantic are partitioned into seven bands between longitudes 250°E, 260°E, 270°E, 280°E, 290°E, 300°E, 310°E and 320°E. Within each band, we determine the average latitude of salinity observations that fall between 34.645 g kg$^{-1}$ and 34.65 g kg$^{-1}$ (using a simple arithmetic mean). This defines a latitude ($y_0(x)$) for each of the seven bands, and effectively determines the approximate path the tracer would follow if it did not mix irreversibly. We use 10° and not narrower bands to avoid noise arising from a minimal number of salinity observations being made on the isopycnal.

A smooth set of points every 0.5° of longitude are then used to define the curve for $y_0(x)$, using a 1-dimensional cubic spline interpolation.

For each isopycnal salinity anomaly observation ($S'$) we determine its distance ($D(S')$) from the curve $y_0(x)$. Distance is defined as the minimum great-circle distance from the observation to a point on the curve $y_0(x)$. The mean distance, $\overline{D(S')}$, is then the average of $D(S')$ for a given band of longitudes and over a range of salinities close to $S'$. $\overline{D(S')}$ is used to transform

the salinity anomaly coordinate into an equivalent distance coordinate. In regions where $y_0(x)$ follows a line of constant latitude (as is approximately the case in the South Pacific), $\overline{D(S')}$ is effectively the average meridional distance of water parcels with salinity S', from the line $y_0(x)$. $D(S')$ is averaged within each 10° band and within salinity bins of 0.005 g kg$^{-1}$, as shown in Fig.7b.

Our distance coordinate is analogous to other pseudo-distances used in diffusivity studies in the Southern Ocean, where

quasi-stream-following coordinates are used. For example, Naveira Garabato et al. (2011) use a similar approximate conversion between dynamic height and latitudinal distance when estimating eddy stirring length scales. It should be noted that there is





some unavoidable loss of spatial information in the use of water mass-following coordinates. While the distance coordinate aims to relate changes in salinity at constant density to changes in distance, it is impossible to know the actual path taken by tracers, and so the distance conversion will always be approximate. We discuss the impact of changes in both the distance coordinate with longitude and its impact on apparent diffusivity in the next section.

The distance $\overline{D(S')}$ decreases substantially between the South Pacific and the Scotia Sea. At the release site in the South Pacific waters become 0.01 g/kg fresher over approximately 700 km, while they change by the same amount over only 300 km in the Scotia Sea.

In order to translate the salinity second moment (Fig. 5) into units of distance squared (as relevant to the estimation of a mixing coefficient) we construct 'reference profiles' based on $\overline{D(S')}$. The use of a reference profile is routine when converting isopycnally averaged tracer observations into vertical distance for the purpose of inferring a vertical diffusivity (Ledwell et al., 1993). Since there is a substantial change in the salinity gradient between the South Pacific and the Scotia Sea, and there is an apparent change in the rate at which the tracer spreads in salinity after 18-24 months, we have defined three mean profiles for: the entire region between 250°E and 310°E, a western region between 250°E and 290°E, and an eastern region between 280°E and 310°E. The choice of longitudes here is arbitrary, and our aim is to merely assess the sensitivity of second moments and diffusivities estimated to a range of possible profiles.

Given distance as a function of salinity for each section, $\overline{D(S')}$, we use the climatological hydrography to map the tracer from isopycnal salinity-anomaly to distance-anomaly coordinates. The same distance versus salinity anomaly profiles are used on all isopycnals.

We apply a non-linear least-squares fit of a two-dimensional Gaussian distribution to each section to yield the standard width in the isopycnal direction ($\sigma_{iso}$), with the resulting along-isopycnal component of the second moments shown in Fig. 8 now in the conventional units of squared metres.

## 5 Interpretation of the isopycnal growth of the tracer patch

Given the change in the isopycnal second moment at each section ($\Delta\sigma_{iso}^2$) and the time between section occupations ($\Delta t$), we can examine whether the tracer patch grows linearly, as would be expected from the ensemble area of a tracer patch with a constant diffusion coefficient ($K_{iso}$), such that

$$\Delta\sigma_{iso}^2 = 2K_{iso}\Delta t., \tag{1}$$

or if there is a temporal change in the growth rate either indicating different phases of the tracer evolution or geographical changes in $K_{iso}$.

In Figure 8a, the second moments of the tracer are plotted as a function of time since release, using the mean salinity versus distance profile for the entire survey region. The classical calculation of the eddy diffusion coefficient based on Lagrangian data statistics was first introduced by Taylor (1922), who showed that under stationary conditions and after multiples of the Lagrangian correlation time, the area of a cloud of particles will grow linearly in time at a rate known as the effective eddy diffu-





**Figure 7.** a) Salinity observations interpolated onto the release isopycnal surface (circles). A spline fit of the mean latitude of salinities between 34.645 g kg$^{-1}$ and 34.65 g kg$^{-1}$ is shown with a black line. b) The mean minimum distance of observations in each salinity class to the black line in 10° longitude bands. c) The mean salinity versus distance profile for the entire survey region and for a western and an eastern regions. In the eastern region insufficient fresher (negative) salinity anomalies are measured to produce a profile there.



**Figure 8.** Isopycnal second moment of the tracer versus time since release, computed by projecting from tracer versus salinity anomaly into tracer versus distance, using a constant distance versus salinity profile. Panel a) uses the profile from salinity data between $250°$E and $310°$E, b) between $250°$E and $290°$E, and c) $280°$E and $310°$E. Only tracer data gathered between those latitudes are shown. Each small marker represents a single member of the bootstrap ensemble, and the larger marker the result of using all the data. The rate of change of the second moment in time is proportional to an apparent diffusivity (equation 1). Contours of apparent diffusivity are shown in black. a) and b) show the case where the growth is linear from the release date, and c) shows an offset of 21 months. Within each panel are shown the distribution functions of diffusivity estimates. In a) and b) these assume constant mixing from the release date. In c) these estimates are based on the slope between the two eastern South Pacific sections and the two Scotia Sea sections.





sivity. In this paper, we describe the linear growth from an infinitesimally small second moment with an "apparent diffusivity".
We next test this simple model of the growth of the tracer patch.

The apparent diffusivity is on the order of 50 m$^2$ s$^{-1}$ in the South Pacific leading up to the eastern South Pacific and Drake Entry sections, and on the order of 400 m$^2$ s$^{-1}$ integrated over both the South Pacific and Scotia Sea regions. Downstream sections such as the Drake Exit exhibit a large range of apparent diffusivities (i.e. when fitting second moment growth all the way back to the origin) from order 200 m$^2$ s$^{-1}$ to order 600 m$^2$ s$^{-1}$, with later diffusivities tending to be larger even though
the tracer has transited through the same geographical domain.

The change in apparent diffusivity between the earlier period and western region, and the later period and eastern region, can be explained by at least three mechanisms: first, the difference in salinity gradient from the eastern to western region could translate into virtual increase of apparent diffusivity as we have not yet accounted for such variations; second, the tracer may enter a region of more vigorous isopycnal mixing in the east; third, the apparent diffusivity is low at early stage after tracer
release because the tracer is still in a slow exponential growth regime, before achieving faster growth after some lag time on the order of months to years (Garrett, 1983) and coincidentally when it enters the eastern region. The later two possibilities are addressed in more detail in the discussion section below.

To ascertain whether the change in apparent diffusivity could be explained by a change in the salinity gradient itself, we estimate a diffusivity in two phases using salinity versus distance profiles for the western and eastern regions separately (shown
in Fig.7). For the western region, we use only tracer measurements made in the first 26 months of the experiment and to the west of 290°E. These estimates assume that mixing is linear from the release date. For the eastern region, we use only measurements made after 20 months and to the east of 280°E. The eastern region estimates are made by taking the difference in the second moment estimates from the South-East Pacific and Drake Entry sections and those measured at least 6 months later at the Drake Exit and North Scotia sections. Although the statistics of the diffusivity estimates are not normal, we estimate the error bounds
based on the $16^{th}$ and $84^{th}$ deciles (equivalent to $\pm$ 1 standard deviation for a normal distribution).

Considering only the sections in the western region in the first 26 months, this analysis yields a diffusivity of 40–100 m$^2$ s$^{-1}$ (45-130 m$^2$ s$^{-1}$ if the final Drake Entry section at 30 months is included). We estimate a diffusivity of 250–870 m$^2$ s$^{-1}$ in the eastern region. This suggests that even when considering a change in the background salinity gradient there is still a substantial increase of diffusivity between the western and eastern region, so something else must explain this increase of diffusivity. We
note, however, that a change in the background salinity gradient might explain some portion of the apparent increase in the tracer migration to different salinities on the isopycnal.

## 6   Discussion

In a previous study, data from DIMES together with output of a numerical model were used to quantify isopycnal mixing in the eastern South Pacific (Tulloch et al., 2014). Those authors estimated a diffusion coefficient using the ensemble area of the tracer
patch and found it to be of order 700 m$^2$ s$^{-1}$. An additional study used float observations to estimate a diffusion coefficient on the order of 800 m$^2$ s$^{-1}$ (LaCasce et al., 2013). Here, we have taken a complementary approach and attempt to quantify





the irreversible mixing of the tracer as it spreads both in the South Pacific and Scotia Sea regions. Using our novel coordinate framework and assuming linear growth from an infinitesimally small patch, we demonstrate that, as the tracer disperses the evolution of the area of the tracer is consistent with an apparent diffusivity of $70 \pm 30$ m$^2$ s$^{-1}$ in the Southeast Pacific,

increasing to $560 \pm 310$ m$^2$ s$^{-1}$ downstream as the tracer enters the Scotia Sea. While our estimate of apparent diffusivity in the Scotia Sea is consistent with Tulloch et al. (2014) and LaCasce et al. (2013), we note that our estimate in the south east Pacific is significantly lower than their estimate.

While we cannot here disentangle the exact reason for such a difference with previous authors for our estimate in the southeast Pacific, we explore whether the increase of dispersion from the Southeast Pacific to the Scotia Sea could be explained

by the time evolution of the regime of dispersion of the tracer patch rather than by a regional difference in conditions. In other words, we explore the possibility that the tracer dispersion is initially very slow, and rapidly increases after some time lag, making a linear growth assumption like we did above inappropriate. Indeed, (Garrett, 1983) discussed that the evolution of a tracer patch (that has already formed an approximately Gaussian distribution; the DIMES tracer was initially dispersed in a 20 km by 20 km cross shape) should have two distinct phases in its evolution: one first phase where the tracer forms streaks, so

where the actual area of the tracer grows slowly; and a second phase after some time lag as the actual area approaches the area of an ellipse surrounding the tracer, the growth rate of the actual area of the tracer then asymptotes toward linear growth.

The time-lag and time evolution of the area of the tracer patch in the two different regimes can be expressed as a function of a number of parameters of the flow field (Garrett, 1983, see the Appendix). By trying to best estimate those parameters and bootstrapping a best fit our observed tracer patch dispersion with Garrett (1983)'s theoretical prediction, we are able to

estimate the isopycnal mixing $K_{iso}$ reached in the phase of linear growth of the tracer patch, as well as the time lag at which this regime is reached (see the Appendix). From the ensemble of solutions, we find that the isopycnal diffusivity lies between 240 m$^2$ s$^{-1}$ and 550 m$^2$ s$^{-1}$ (Fig.9a shows the distribution). The lag times are between 20 and 32 months. If we use the much lower small-scale diffusivity ($K_s$=0.04 m$^2$ s$^{-1}$; see appendix), the solutions for $K_{iso}$ and the lag time do not change substantially ($K_{iso}$=290–600 m$^2$ s$^{-1}$; lag time = 19-29 months). Both the results of Tulloch et al. (2014) and LaCasce et al.

(2013) are consistent with the linear growth rate that we find here of 240–550m$^2$ s$^{-1}$. This linear growth sets in after an initial slow growth phase, as predicted by the theory of Garrett (1983) and suggested our observation of a step change in diffusivities between the South Pacific and Scotia Sea. Additionally the lag time is consistent with the point at which we note the change in diffusivities.

Alternative explanations for the spread of apparent isopycnal diffusivities in Fig. 8 can not be ruled out. It is possible that

frontal suppression inhibited mixing in the South Pacific and that this broke down as the tracer entered Drake Passage, although this interpretation would arguably be at odds with the diffusivity estimates made by Tulloch et al. (2014) and LaCasce et al. (2013). Alternatively, the tracer growth may not have reached the lag time, and may still be in a relatively weak mixing regime (potentially explaining why the implied diffusivities are on the low side of Tulloch et al. (2014) and LaCasce et al. (2013)). In this case, it may be that geographical changes in vertical mixing drive geographical changes in the size of filaments, and

therefore the apparent isopycnal mixing inferred (Smith and Ferrari, 2009). Vertical mixing was indeed found to increase as the tracer moved from the Pacific Ocean into the Scotia Sea (Watson et al., 2013).





## 7  Conclusions

Observations of a passive tracer released in the South Pacific Ocean have been discussed and we have projected tracer observations onto isopycnal salinity anomaly coordinates. The proposed coordinate is equivalent to isopycnal temperature anomaly

and is stream-following. For the tracer to deviate from a line of constant temperature and salinity, irreversible transformations must occur. Spreading of the tracer in this coordinate hence relates to irreversible mixing.

The second moment of the tracer in salinity coordinates grows initially to order $0.05$ g$^2$/kg$^2$ in the South Pacific over the first 20-25 months. It then grows to order $0.4$ g$^2$/kg$^2$ by 50 months, suggesting a change in the rate at which the tracer spreads through isopycnal salinity space.

In order to relate dispersion in salinity coordinates to isopycnal mixing in geographical coordinates, we have related isopycnal salinity to cross-stream distance in the same way that density has been related to depth in previous studies. The growth of the second moment of the tracer in these equivalent geographical coordinates has then been estimated.

In distance coordinates, the isopycnal second moment grows slowly in the first 26 months at order 75 m$^2$ s$^{-1}$, then rapidly thereafter at order 550 m$^2$ s$^{-1}$. The initial slow growth of the area of a tracer patch proposed by Garrett (1983) is able to

explain these two regimes, although does not entirely preclude alternative explanations. Based on these data, the predicted lag time before the onset of linear growth of the tracer patch area is 20–32 months. The rate of isopycnal mixing (representative of the large-scale dispersion of the tracer) is predicted to be 240–550m$^2$s$^{-1}$ and is consistent with, although somewhat lower than, two recent studies of the same region and period.

## Appendix A

According to Garrett (1983), a tracer patch that has already formed an approximately Gaussian distribution (the DIMES tracer was initially dispersed in a 20 km by 20 km cross shape) should have two distinct phases in its evolution:

1. The tracer forms streaks. The area of an ellipse surrounding the tracer grows much faster than the actual area of the tracer (A), which evolves according to

$$A = \pi \frac{K_s}{\gamma} e^{\alpha\gamma\left(t - \frac{1}{4}\gamma^{-1}\right)}, \tag{A1}$$

where $K_s$ is a small-scale diffusivity, $\gamma = \sqrt{u_x^2 + v_y^2}$ is the strain rate ($u_x$ is the zonal gradient of the zonal velocity, and $v_y$ is the meridional gradient of the meridional velocity), and $\alpha$ is a constant of order 1.

2. As the actual area approaches the area of an ellipse surrounding the tracer, the growth rate of the actual area of the tracer then asymptotes toward linear growth, where $K_{iso}$ obeys (Eq. 1). We term the time at which (1) and (A1) give the same value for $A$ and $\sigma_{iso}$ the *lag time* (specifically referred to as $t_2$ in Garrett (1983)).

Using typical values for the stratification and strain in the ocean, Garrett (1983) estimated the lag time to be on the order of 1 year. To complete this calculation for the region in question, the small-scale diffusivity ($K_s$), strain ($\gamma$), and $\alpha$ are required.

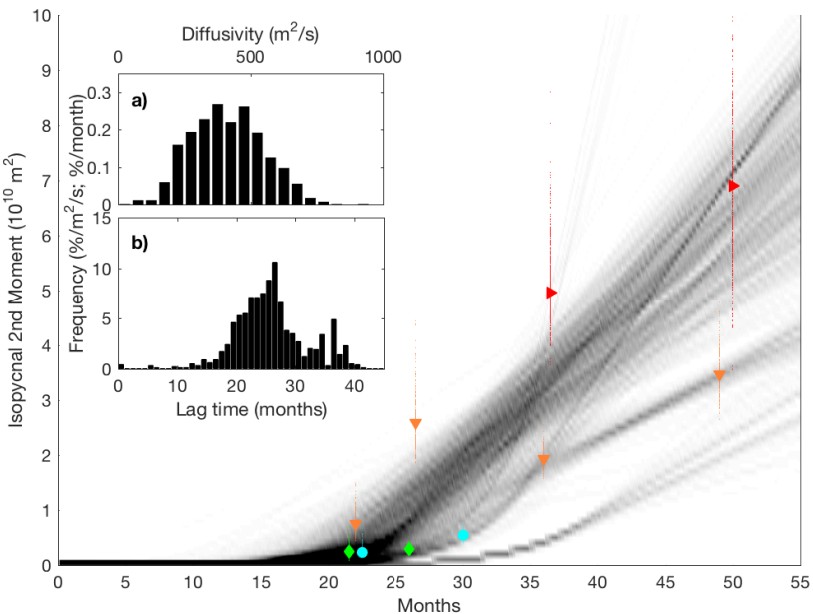

**Figure A1.** As in Fig. 8, but with the shading showing the density of solutions from 1000 best fits of the Garrett (1983) model where the sections are randomly sub-sampled. The two subplots show the distribution of isopycnal mixing coefficients (a) and lag times (b).

We use the small-scale diffusivity estimated by Boland et al. (2015) of 20 m$^2$ s$^{-1}$. As noted by Boland et al. (2015), this value is about three orders of magnitude larger than computed from the equation proposed by Young et al. (1982) and used in the Garrett (1983) study. It is also one order of magnitude larger than the 2-3m$^2$ s$^{-1}$ estimated by (Ledwell et al., 1993) for a tracer

patch at 300 m depth in the eastern North Atlantic. We therefore test the sensitivity of the lag time to this choice below. To determine the strain ($\gamma$), we use SatGEM data (Meijers et al., 2011) mapped onto the release isopycnal and interpolated onto the $y(\overline{S})$ line defined in section 4 based on satellite and hydrographic data from the DIMES survey period. Between 250°E and 290°E, we find $\gamma \approx 2 \times 10^{-6} s^{-1}$.

The most significant uncertainty in the estimation of the lag time lies in our choice of $\alpha$. As most of the variation from site to

site of the dispersion process is presumably captured by $\gamma$, one might expect $\alpha$ to be roughly the same at each site of mesoscale stirring. The variation of $\alpha$ would quantify changes in the statistics of the strain other than its variance, for example variations in the Lagrangian autocorrelation of the strain rate.

Given the above numbers and using $\alpha = 0.5$ (the value adopted by Garrett (1983)) we obtain a lag time of 2 months. With $\alpha = 0.2$ (the value estimated from a simulated tracer release in the North Atlantic by Lee et al. (2009)), the lag time increases

to 6 months, and for $\alpha = 0.1$ and 0.05 the lag times are 13 and 29 months, respectively. We estimate the uncertainty in $\gamma$ to be 10-20%, by shifting the averaging window for the SatGEM fields by 10° to the east and west. Although the strain rate could be





diagnosed in a range of ways, potentially leading to a larger uncertainty, it is likely that $\gamma$ has less influence on the uncertainty in (A1) than $\alpha$.

As the lag time is uncertain, we choose to use equation (1) to solve for both $K_{iso}$ and $\alpha$ using our estimates of $\sigma_{iso}^2$. In

order to capture the behaviour of slow followed by fast growth phases, we match (1) to (A1) for the slow growth period with $\sigma_{iso}^2 = A/2\pi$. Implicit in this fitting procedure is the assumption that, during the initial growth phase, our second moments, based on the tracer projected from salinity to distance coordinates, are related to the total area of the tracer patch. We cannot support this with theory. We simply use equation (A1) as a model that offers the behaviour of slow followed by fast growth. We only interpret the inferred lag time and $K_{iso}$ that describe the linear growth phase.

Using the entire ensemble of $\sigma_{iso}^2$ estimates, we solve for $K_{iso}$ and $\alpha$ such that the continuous line matching (A1) and (1) at the lag time gives the best fit to the estimated second moments (using Matlab's fminsearch algorithm). We estimate the uncertainty again by bootstrapping, this time subsampling the 11 sections entering the estimate. That is, for each ensemble member, 11 non-unique random numbers are chosen and the corresponding section data are used to estimate $K_{iso}$ and $\alpha$. This process is repeated 1000 times to form an ensemble of $K_{iso}$ and $\alpha$ estimates.

*Competing interests.* The authors have no competing interests.

*Acknowledgements.* This research was supported by the Natural Environment Research Council (NERC), and the DIMES project was also supported by the National Science Foundation (NSF). We thank Jim Ledwell for many thoughtful contributions and discussions regarding this manuscript. We also thank all those who worked in the preparation and execution of the DIMES experiment. The data are publicly available through the British Oceanographic Data Centre (www.bodc.ac.uk/projects/data_management/international/dimes/).





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
