# Peer review of "Tracking the spread of a passive tracer through Southern Ocean water masses"

_Ocean Science, 2019_

## Referee Comment (RC1) · Anonymous Referee #1 · 30 Sep 2019

In 2009, as part of the Diapycnal and Isopycnal Mixing Experiment in the Southern Ocean (DIMES), a passive tracer was released near 1500 m depth upstream of Drake Passage. Subsequent cruises measured tracer concentrations across the Antarctic Circumpolar Current to document the gradual tracer spreading due to mixing and advection. The tracer data has been analyzed in density space and geographical space to derive integral constraints on levels of irreversible mixing and lateral stirring (Watson et al. 2013, Tulloch et al. 2014). Here, the authors use an original approach: they analyze tracer spreading along isopycnals but across isohalines. The use of salinity as the lateral coordinate yields new insights about Southern Ocean stirring and mixing processes. The analysis is carefully exposed and the results well presented. Nonetheless, the manuscript could improve with a more pedagogical explanation of the rationale and

of the hypotheses discussed. I detail this concern below.

Main comments:

1. The important distinction between reversible (lateral) stirring and irreversible (isotropic) mixing is not clearly explained and is only discussed in two sentences in the discussion (lines 284-286). I think this point deserves more space throughout the manuscript. Few readers will be as familiar as the authors with this distinction.

You make the point in the introduction that spreading across isohalines implies irreversible mixing. Then you move on to analyze tracer spreading in isopycnal salinity coordinates in terms of lateral diffusivities, leaving out isotropic (small-scale) diffusivities (until lines 284-286). By focusing on lateral diffusivities, you make the implicit assumption that they play the most important role in shaping the ultimate irreversible mixing. Yet irreversible mixing depends on the interaction of lateral stirring (which produces gradients) with isotropic mixing (which consumes gradients); it is not obvious which of the two processes should be the dominant cause of variations in spreading rate.

Actually, to me, the 20-fold increase in isotropic mixing from west to east of Drake Passage (Watson et al. 2013) would be the most natural 'default' hypothesis for the inferred increase in spreading rate in your isopycnal salinity coordinate system. Could you discuss why you expect along-stream changes in lateral stirring rate to be more important?

More generally, could you make clear upfront that isopycnal stirring alone cannot mix the tracer across isohalines? Is it possible to place constraints on both reversible stirring and irreversible mixing via your analysis? Could the difference between the isopycnal diffusivities you infer and previous estimates (Tulloch et al. 2014, LaCasce et al. 2013) relate to the influence of isotropic diffusivities on spreading within your framework?

2. Lines 123-125 you state that the reference salinity varies from section to section "to account for the migration of the peak due to gradual along-isopycnal transport". This explanation is unclear to me. Do you mean that the peak slowly erodes due to small-scale isotropic mixing? Could meaningful information about isotropic mixing be hidden in this variation of the peak salinity? Should the reference salinity also vary with density within each section?

3. It would really help to have Figure 7a at the beginning of the manuscript. The salinity ridge is key to the overall analysis.

4. Why do you choose salinity rather than temperature as your cross-stream variable?

5. Does isopycnal mixing in the longitudinal direction matter for your analysis?

Specific comments:

Line 39: "a water parcel in the interior"

Figure 3: Mention changes in vertical scale between panels in legend.

Line 149: delete "the" at end of line.

Lines 173: "if it did not mix irreversibly", it would not undergo the decrease in salinity observed along $y_0(x)$.

Legend of figure 7: last sentence does not seem to be correct.

Legend of figure 8: "between those latitudes": I think you mean "between those longitudes".

Lines 244-246: I don't understand this point: please clarify.

---

## Referee Comment (RC2) · Anonymous Referee #2 · 8 Nov 2019

**Overview**

This paper estimates meridonal mixing of a passive tracer released in the Southern ocean, and tracked over ∼3 years. Previous analyses from the same experiment measured mixing via conventional x-y (longitude-latitude) distance coordinates. The present work utilizes salinity variations and large-scale salinity gradients on isopycnals to infer irreversible mixing rates of the tracer in salinity space, and then relates this back to physical space spreading via climatological salinity gradients in the region. Results are roughly consistent with previous published results based on traditional second-moment diffusivity calculations, as well as diffusivities inferred from floats.

**General Comments**

[Figure]

The authors present an interesting approach for inferring irreversible mixing on isopycnals, arguing that analysis in salinity space along isopycnals provides a measure of irreversible mixing as the tracer advects through varying meridonal salinity gradients. The rationale for the present analysis is that as the meridional gradient of salinity changes along the path of mean tracer advection due to meridonal convergences and divergences, which in physical space cause the tracer to widen, then narrow, then widen again as it is transported zonally. The authors liken the analysis in salinity space along isopycnals to analysis in density space when measuring diapycnal mixing, the advantage of the latter being that it enables the separation of spreading by isopycnal straining from that caused by irreversible diapycnal mixing.

While the approach will readily make sense to those familiar with this type of tracer analysis, I share the other reviewer's sentiment that there are a number of aspects of the method that could benefit from further discussion. First, when comparing the salinity coordinate approach to the isopycnal coordinate approach used to measure diapycnal mixing, the latter is used to account for spatio-temporal trends in stratification as well as high frequency variations from one profile to the next caused by internal waves. While the salinity method certainly addresses larger-scale trends in the meridional salinity gradient, it is not clear whether there are also smaller-scale variations, e.g., due to smaller-scale frontal meanders within the ACC or any of its smaller fronts? If so, this would be useful to point out as one of the distinctions between the salinity-space derived estimate of Kh and other estimates. Also regarding the larger spatio-temporal trends, it would appear that the salinity-space approach is directly analogous to accounting for the convergence and then divergence of the flow as it traverses the Drake passage. This then can be likened to the strain-diffusion balance that is often used to estimate small-scale diffusivity either from the streaking phase of large-scale tracer experiments, or for smaller shorter-time scale fluorescent dye experiments. Effectively, diffusivity estimated in stream-wise coordinates, allowing for converging and diverging strain, should yield similar results. Finally, another difference between the present salinity coordinate analysis and isopycnal coordinate analysis is that the latter

typically is converted back to physical space coordinates using the mean density-depth relation computed from the same data. The use here of climatological salinity data presumes that the salinity gradient during DIMES was similar to the climatological gradient computed from all previous records. While this may be a reasonable (even if necessary) assumption, and any differences likely to be small, a sentence mentioning this as another source of error is warranted.

A second aspect of the results that could benefit from further discussion are some of the nuances of the difference between this salinity space estimate of Kh and other physical space estimates, particular as relates to the difference between mixing and stirring. The authors argue that the present analysis measures "irreversible mixing". By this, they strictly mean mixing that crosses isohalines. However, in practice, by virtue of its random nature, stirring by mesoscale eddies is also irreversible. Beyond semantics, one can consider the phases of stirring and mixing described by Garrett (1983) and cited in the paper. During the early stirring phase, eddies re-distribute the tracer and increase is variance in x-y space, but do little to it in salinity space (e.g., per Section 3, line 111 of the paper). The salinity space "diffusion" occurs due to small-scale diffusive processes that rectify the stirring motions by smoothing out the wisps and streaks across what are then also wisps and streaks in salinity. Once the tracer has begun to fill in across many eddy stirring events, the large-scale variance approaches its linear eddy diffusivity growth regime, and absent spatio-temporal changes in the large-scale salinity gradient, this result should be similar to a physical space analysis of diffusivity. However, before the tracer has filled in between the streaks, wouldn't the salinity space vs. physical space dispersion estimates be expected to differ significantly? What new information about stirring vs. mixing can be gleaned from this? Some clarification would help the reader understand what the salinity space analysis is telling us for these early vs. late times in the tracer evolution.

Specific Comments (Relating to above General Comments)

Line 30: But isn't rearrangement by mesoscale flows, if they are random and/or in

practice do not reverse, what we consider mesoscale eddy stirring, which is its own form of mixing?

Lines 101-102: This is not a problem if one knows the meridonal convergence or strain rate. If this is known, the changing width of the patch in spite of this strain can be computed, which is presumably what the analysis in salinity space will facilitate.

Lines 111-112: "... preserve their temperature and salinity values" ... Except for mixing along mixing lines in T-S space?

Lines 126-128: "... in density versus salinity anomaly coordinate the tracer spreads out more monotonically." ... As it must, since there is no way to mix to different salinities except along mixing lines.

Lines 213-214: Is there a way to assess whether increased diffusive flux is due to greater diffusivity or larger tracer gradient caused by flow convergence?

Review Summary and Rating

Overall I find this paper interesting and worthy of publication in Ocean Science. I have noted above a few points that the authors might consider adding to the Discussion of the paper – among these are some things that would help clarify the analogies between the present approach and previous ones, and also things that might help readers better understand the differences between the present diffusivity estimates and more traditional physical space estimates.

---

## Author Comment (AC1) · 10 Dec 2019

In 2009, as part of the Diapycnal and Isopycnal Mixing Experiment in the Southern Ocean (DIMES), a passive tracer was released near 1500 m depth upstream of Drake Passage. Subsequent cruises measured tracer concentrations across the Antarctic Circumpolar Current to document the gradual tracer spreading due to mixing and advection. The tracer data has been analyzed in density space and geographical space to derive integral constraints on levels of irreversible mixing and lateral stirring (Watson et al. 2013, Tulloch et al. 2014). Here, the authors use an original approach: they analyze tracer spreading along isopycnals but across isohalines. The use of salinity as the

lateral coordinate yields new insights about Southern Ocean stirring and mixing processes. The analysis is carefully exposed and the results well presented. Nonetheless, the manuscript could improve with a more pedagogical explanation of the rationale and of the hypotheses discussed. I detail this concern below.

> We thank the reviewer for these encouraging comments.

Main comments: 1. The important distinction between reversible (lateral) stirring and irreversible (isotropic) mixing is not clearly explained and is only discussed in two sentences in the discussion (lines 284-286). I think this point deserves more space throughout the manuscript. Few readers will be as familiar as the authors with this distinction. You make the point in the introduction that spreading across isohalines implies irreversible mixing. Then you move on to analyze tracer spreading in isopycnal salinity coordinates in terms of lateral diffusivities, leaving out isotropic (small-scale) diffusivities (until lines 284-286). By focusing on lateral diffusivities, you make the implicit assumption that they play the most important role in shaping the ultimate irreversible mixing. Yet irreversible mixing depends on the interaction of lateral stirring (which produces gradients) with isotropic mixing (which consumes gradients); it is not obvious which of the two processes should be the dominant cause of variations in spreading rate.

> Indeed. We have added an important paragraph to the introduction on lines 29-39 with regard to the above points.

Actually, to me, the 20-fold increase in isotropic mixing from west to east of Drake Passage (Watson et al. 2013) would be the most natural 'default' hypothesis for the inferred increase in spreading rate in your isopycnal salinity coordinate system. Could you discuss why you expect along-stream changes in lateral stirring rate to be more important?

> We have further clarified why we have given more weight to the Garrett 1983 model on lines 293-294. The main point here is that even in the absence of changes in stirring

or small scale mixing, we would expect an uptick in the rate of irreversible mixing simple due to the expected time evolution of the tracer patch.

More generally, could you make clear upfront that isopycnal stirring alone cannot mix the tracer across isohalines? Is it possible to place constraints on both reversible stirring and irreversible mixing via your analysis? Could the difference between the isopycnal diffusivities you infer and previous estimates (Tulloch et al. 2014, LaCasce et al. 2013) relate to the influence of isotropic diffusivities on spreading within your framework?

> We have added this statement on lines 58-59. We are unable to delve much further into the differences in reversible verses irreversible stirring, mostly because, as Garrett argued, the rate of stirring and the rate of irreversible mixing will converge over a year or two.

2. Lines 123-125 you state that the reference salinity varies from section to section "to account for the migration of the peak due to gradual along-isopycnal transport". This explanation is unclear to me. Do you mean that the peak slowly erodes due to small-scale isotropic mixing? Could meaningful information about isotropic mixing be hidden in this variation of the peak salinity? Should the reference salinity also vary with density within each section?

> We have reworded those two paragraphs (lines 131-139) which we hope clarifies the matter. Unfortunately the signal isn't sufficiently clear to discern anything robust.

3. It would really help to have Figure 7a at the beginning of the manuscript. The salinity ridge is key to the overall analysis.

> We agree and have included this panel in Figure 1 instead.

4. Why do you choose salinity rather than temperature as your cross-stream variable?

> We now point out on lines 56-58 that they are in fact both equivalent.

5. Does isopycnal mixing in the longitudinal direction matter for your analysis?

> It could. We have added a caveat regarding this to the discussion on lines 303-307.

Specific comments:

Line 39: "a water parcel in the interior"

> Changed

Figure 3: Mention changes in vertical scale between panels in legend.

> Done

Line 149: delete "the" at end of line.

> Deleted

Lines 173: "if it did not mix irreversibly", it would not undergo the decrease in salinity observed along $y_0(x)$.

> Some observations of salinities between the range 34.645 g kg$^{-1}$ and 34.65 g kg$^{-1}$ are observed all the way along the path.

Legend of figure 7: last sentence does not seem to be correct.

> Thank you. Yes, we had that back to front.

Legend of figure 8: "between those latitudes": I think you mean "between those longitudes".

> Thank you again.

Lines 244-246: I don't understand this point: please clarify.

> We have deleted this sentence as ultimately it was not relevant.

---

## Author Comment (AC2) · 10 Dec 2019

Overview This paper estimates meridonal mixing of a passive tracer released in the Southern ocean, and tracked over âĹij3 years. Previous analyses from the same experiment measured mixing via conventional x-y (longitude-latitude) distance coordinates. The present work utilizes salinity variations and large-scale salinity gradients on isopycnals to infer irreversible mixing rates of the tracer in salinity space, and then relates this back to physical space spreading via climatological salinity gradients in the region. Results are roughly consistent with previous published results based on traditional second- moment diffusivity calculations, as well as diffusivities inferred from floats.

[Figure]

> We thank the reviewer for these encouraging comments.

General Comments The authors present an interesting approach for inferring irreversible mixing on isopycnals, arguing that analysis in salinity space along isopycnals provides a measure of irreversible mixing as the tracer advects through varying meridonal salinity gradients. The rationale for the present analysis is that as the meridional gradient of salinity changes along the path of mean tracer advection due to meridonal convergences and divergences, which in physical space cause the tracer to widen, then narrow, then widen again as it is transported zonally. The authors liken the analysis in salinity space along isopycnals to analysis in density space when measuring diapycnal mixing, the advantage of the latter being that it enables the separation of spreading by isopycnal straining from that caused by irreversible diapycnal mixing.

While the approach will readily make sense to those familiar with this type of tracer analysis, I share the other reviewer's sentiment that there are a number of aspects of the method that could benefit from further discussion. First, when comparing the salinity coordinate approach to the isopycnal coordinate approach used to measure diapycnal mixing, the latter is used to account for spatio-temporal trends in stratification as well as high frequency variations from one profile to the next caused by internal waves. While the salinity method certainly addresses larger-scale trends in the meridional salinity gradient, it is not clear whether there are also smaller-scale variations, e.g., due to smaller-scale frontal meanders within the ACC or any of its smaller fronts? If so, this would be useful to point out as one of the distinctions between the salinity-space derived estimate of Kh and other estimates.

> Yes this is an important distinction. We have discussed this more deeply on lines 29-39.

Also regarding the larger spatio-temporal trends, it would appear that the salinity-space approach is directly analogous to accounting for the convergence and then divergence of the flow as it traverses the Drake passage. This then can be likened to the straindiffusion balance that is often used to estimate small-scale diffusivity either from the streaking phase of large-scale tracer experiments, or for smaller shorter-time scale fluorescent dye experiments. Effectively, diffusivity estimated in stream-wise coordinates, allowing for converging and diverging strain, should yield similar results.

> Yes this is a good point. In this case we are fortunate to have temperature and salinity measured at the same bottles as the tracer. Unfortunately, we do not have estimates of streamline positions relevant to the particular depth/isopycnal of the tracer.

Finally, another difference between the present salinity coordinate analysis and isopycnal coordinate analysis is that the latter typically is converted back to physical space coordinates using the mean density-depth relation computed from the same data. The use here of climatological salinity data presumes that the salinity gradient during DIMES was similar to the climatological gradient computed from all previous records. While this may be a reasonable (even if necessary) assumption, and any differences likely to be small, a sentence mentioning this as another source of error is warranted.

> Indeed, we have added this statement to lines 172-174.

A second aspect of the results that could benefit from further discussion are some of the nuances of the difference between this salinity space estimate of Kh and other physical space estimates, particular as relates to the difference between mixing and stirring. The authors argue that the present analysis measures "irreversible mixing". By this, they strictly mean mixing that crosses isohalines. However, in practice, by virtue of its random nature, stirring by mesoscale eddies is also irreversible. Beyond semantics, one can consider the phases of stirring and mixing described by Garrett (1983) and cited in the paper. During the early stirring phase, eddies re-distribute the tracer and increase is variance in x-y space, but do little to it in salinity space (e.g., per Section 3, line 111 of the paper). The salinity space "diffusion" occurs due to small-scale diffusive processes that rectify the stirring motions by smoothing out the wisps and streaks across what are then also wisps and streaks in salinity. Once the tracer has begun to

fill in across many eddy stirring events, the large-scale variance approaches its linear eddy diffusivity growth regime, and absent spatio-temporal changes in the large-scale salinity gradient, this result should be similar to a physical space analysis of diffusivity. However, before the tracer has filled in between the streaks, wouldn't the salinity space vs. physical space dispersion estimates be expected to differ significantly?

> Yes indeed. Although we have not adequately communicated so, this is our principle explanation for the behaviour of the tracer. We have improved the discussion of stirring versus mixing in the introduction including further reference to Garrett (lines 29-39) and tried to make our explanation more clear in the discussion section (lines 292-294).

What new information about stirring vs. mixing can be gleaned from this? Some clarification would help the reader understand what the salinity space analysis is telling us for these early vs. late times in the tracer evolution.

> Basically our main conclusions are that if you are looking for a tracer in the ocean the best place to look is at the density and salinity and temperature you released it at. And secondly that Garrett was probably right. We would like to say more quantitative things but these data alone will not allow it.

Specific Comments (Relating to above General Comments) Line 30: But isn't rearrangement by mesoscale flows, if they are random and/or in practice do not reverse, what we consider mesoscale eddy stirring, which is its own form of mixing?

> We agree this was unclear and have reworded.

Lines 101-102: This is not a problem if one knows the meridonal convergence or strain rate. If this is known, the changing width of the patch in spite of this strain can be computed, which is presumably what the analysis in salinity space will facilitate.

> Yes, but we do not have accurate strain data flowing the tracer.

Lines 111-112: ". . . preserve their temperature and salinity values" . . . Except for mixing along mixing lines in T-S space?

> We still think this statement holds as we are making the point that filaments are largely adiabatic and feel that mentioning T-S lines would be a distraction. We have added the term 'largely'.

Lines 126-128: ". . . in density versus salinity anomaly coordinate the tracer spreads out more monotonically." . . . As it must, since there is no way to mix to different salinities except along mixing lines.

> While this is strictly true if we observed the entire tracer, but since we are only observing individual sections we would prefer to leave this as an observation rather than a statement of fact.

Lines 213-214: Is there a way to assess whether increased diffusive flux is due to greater diffusivity or larger tracer gradient caused by flow convergence?

> In a qualitative sense we have done this by showing how the diffusivity estimates change with different salinity vs distance profiles but we are not comfortable saying too much more.

Review Summary and Rating Overall I find this paper interesting and worthy of publication in Ocean Science. I have noted above a few points that the authors might consider adding to the Discussion of the paper – among these are some things that would help clarify the analogies between the present approach and previous ones, and also things that might help readers better understand the differences between the present diffusivity estimates and more traditional physical space estimates.

> Thank you again for these helpful suggestions and encouragement.

---

## Author Response (AR2)

Response to reviewer comments on "Tracking the spread of a passive tracer through Southern Ocean water masses".

**Anonymous Referee #1**

The revisions made address my previous comments.

One remaining comment:

In the revised manuscript, the favored explanation for the change in isopycnal mixing rate with time is the regime shift predicted by Garrett (1983). Oddly, the abstract does not mention this explanation; rather, it emphasizes the other two possibilities (changes in background gradient or in background mixing). The same holds for the sentence line 166-167.

Good point. We agree that the explanation in the abstract does not fit well with the actual conclusions of the paper. We have changed the text to read

"…while part may be explained by the evolution of the tracer patch from a slowly growing phase where the tracer forms filaments to a more rapid phase where the tracer mixes at 240–550m$^2$s$^{-1}$."

And line 165-167 to read

"This could be explained by the natural phases of evolution of the tracer patch, a change in the rate at which the tracer is mixed, or by a change in background salinity gradient."

One typo: line 286, "suggested by".

Changed

**Anonymous Referee #2**

The authors have sufficiently addressed my previous comments. However, on re-reading of the manuscript, there are still some points of potential confusion the authors may wish to address in terms of how they describe the application of the Garrett (1983) model to the present data. Specifically, there are a number of places in the manuscript where the authors refer to Garrett's 2nd and 3rd phases of tracer dispersal (the streakiness and eddy diffusivity phases, which are the present manuscript's 1st and 2nd phases) as "slow" and then "fast" respectively. One point of potential confusion with this is that one typically thinks of exponential growth as faster than linear growth, such that we might expect dispersion during the streaky phase to be faster than during the linear phase. When considering the two spreading phases in the context of diffusivity, however, the real quantity of interest is the time rate of change of the area. Comparing an exponential to linear growth rate in tracer area, linear growth will always be faster than exponential growth for small times. However, eventually, an exponential growth rate (which is also an exponentially growing diffusivity) will always overtake a linear growth rate (i.e., constant diffusivity). In terms of tracer patch areas, as I understand it, the theoretical time in Garrett's analysis referred to in the appendix as the "lag time" is the cross-over point when an exponential growth in area (not diffusivity) will overtake a linear growth in area for the same initial condition.

Considering the slopes of the linear and exponential area growth curves, i.e., the diffusivity, however, the slope of the linear growth curve is constant (i.e., constant diffusivity) for all time, while the slope of the exponential curve is itself exponential in time. Thus the slope of the exponential growth increases from a small value to large value with time. This then gets back to the point of potential confusion – the slope of the exponential growth phase, when interpreted as a diffusivity, starts out very small, then becomes large, at first being smaller than the linear growth rate (i.e., K_iso), then becoming larger, but both of these occur before the "lag time" is reached, i.e., both during the exponential growth phase of the tracer. Only if one samples early in the exponential growth phase would one expect to see a smaller slope, and hence infer a smaller diffusivity than K_iso. How early is early could be computed, but I believe it is different from Garrett's 1/(4*\gamma). This should be clarified in any discussion of smaller inferred diffusivities during the exponential growth phase vs. the later linear growth phase.

Indeed we feel that most readers will see the term 'exponential' to describe a slower growth phase is confusing. For this reason we now emphasise only the 'slow phase' rather and do not mention the 'exponential' phase at all. E.g. on line 245:

"the apparent diffusivity is low at early stage after tracer release because the tracer is still growing slowly as it forms filaments, before achieving faster growth"

The above points are confused even further when one considers that Garrett's model addresses diffusivity defined in terms of growth of tracer area. Not clear is how the exponential vs. linear growth phases behave when considered in a large-scale natural tracer coordinate system as discussed in this paper, where both the anthropogenic tracer and the large-scale passive tracer are being advected and diffused. In that case, are there still slow and fast phases of the exponential growth, even though diffusion across isopycnal salinity contours is driven solely by small-scale diffusive processes, and not along-streak straining? This aspect likely gets into greater detail than is warranted in the present manuscript. However, it reinforces the point about being careful in how one describes the initial "slow" growth of tracer, despite this being during the exponential growth phase.

Indeed this is an important limitation however we feel the approach we have taken to estimate the area and its caveats is discussed thoroughly (e.g. line 43 and 183).

Lines 147-149, and Fig 5: Relative to Fig. 5 and associated analysis, if the ocean is also stratified vertically via salinity, then wouldn't diapycnal mixing alone show a spread in both density and salinity in these pictures? By measuring the 2nd moment of spread in salinity only, and attributing this all to isopycnal mixing, doesn't this incorrectly include effects of diapyncal mixing? Put another way, for diapynal mixing in a salinity stratified ocean, wouldn't the ellipses in Fig. 4 be tilted, with a covariance between density and salinity? If so, this effect would need to be accounted for in order to infer isopycnal mixing across lateral salinity gradients. It would seem that the possible importance of this could be easily estimated by considering the diapyncal mixing rate applied to the vertical salinity gradient compared to the isopycynal mixing rate applied to the lateral salinity gradient to see which dominates the change in salinity of the tracer.

Salinity anomaly is a function of density. That is, for each density a salinity anomaly is relative to a different reference value. This is articulated on lines 134-137.

Line 285: The statement that the actual area of tracer grows slowly in the "first phase" described here (Garrett's 2nd phase) is confusing for the reasons noted above - the streaky phase represents first a slow then a rapid growth rate (i.e., diffusivity), while the eddy stirring phase is a constant growth rate (i.e., diffusivity).

Line 296: Again, I would not call this a "slow growth" phase, as it is exponential growth of tracer area, first slow then fast, as opposed to linear growth (constant growth rate) that happens later in the eddy stirring regime.

We feel that emphasising, as we do here, that the initial phase is where filaments are formed is clear. Indeed if we emphasised that the evolution was exponential (which we now don't mention in this Discussion) we feel this would confusion.

Not with respect to any particular passage, but relative to the fitting of Garrett's model to a large-scale multi-year tracer experiment, in addition to Ledwell et al (1993) and Ledwell et al (1998), it might also help for context to cite the analysis of Sundermeyer and Price (1998) as applied to the North Atlantic Tracer Release Experiment. In particular, note Figs. 2 and 10 in the latter showing observed and modeled tracer moments relative to Garrett's exponential and eddy diffusion growth phases.

Thankyou. This was indeed an important reference and has been added in the introduction.

[revised manuscript text omitted]